# Large Scale Synthesis of Nanopyramidal-Like VO_2_ Films by an Oxygen-Assisted Etching Growth Method with Significantly Enhanced Field Emission Properties

**DOI:** 10.3390/nano9040549

**Published:** 2019-04-04

**Authors:** Zongtao Zhang, Yifei Feng, Yanfeng Gao, Deliang Chen, Guosheng Shao

**Affiliations:** 1School of Materials Science and Engineering, Zhengzhou University, Zhengzhou 450001, China; yffeng@163.com (Y.F.); gsshao@zzu.edu.cn (G.S.); 2School of Materials Science and Engineering, Shanghai University, Shanghai 200444, China

**Keywords:** vanadium oxide, nanopyramid, metal-insulator phase transition, oxygen etching, field-emission property

## Abstract

The present investigation reported on a novel oxygen-assisted etching growth method that can directly transform wafer-scale plain VO_2_ thin films into pyramidal-like VO_2_ nanostructures with highly improved field-emission properties. The oxygen applied during annealing played a key role in the formation of the special pyramidal-like structures by introducing thin oxygen-rich transition layers on the top surfaces of the VO_2_ crystals. An etching related growth and transformation mechanism for the synthesis of nanopyramidal films was proposed. Structural characterizations confirmed the formation of a composite VO_2_ structure of monoclinic M1 (P21/c) and Mott insulating M2 (C2/m) phases for the films at room temperature. Moreover, by varying the oxygen concentration, the nanocrystal morphology of the VO_2_ films could be tuned, ranging over pyramidal, dot, and/or twin structures. These nanopyramidal VO_2_ films showed potential benefits for application such as temperature−regulated field emission devices. For one typical sample deposited on a 3-inch silicon substrate, its emission current (measured at 6 V/μm) increased by about 1000 times after the oxygen-etching treatment, and the field enhancement factor β reached as high as 3810 and 1620 for the M and R states, respectively. The simple method reported in the present study may provide a protocol for building a variety of large interesting surfaces for VO_2_-based device applications.

## 1. Introduction

Structural characteristics including dimensionality, morphology, and crystal structure can have significant influences on the properties and applications of functional materials. The development of rational processes for synthesizing materials with tunable micro-/nanostructures is essential for both fundamental research and industrial applications. Multivalent transition metal oxides, e.g., TiO_x_, VO_x_, WO_x_, and NiO_x_, usually exhibit a wealth of interesting and useful properties that are beyond that of conventional semiconductors [1,2], due to their abounding types of order in the spin, charge, or orbital degree of freedom, which are associated with the convertible valence states of metal cations [3]. Among these oxides, vanadium dioxide (VO_2_), a prototypical strongly correlated electron oxide, has drawn growing attention because of its famous reversible first-order metal-to-insulator phase transition (MIPT), which occurs at an accessible temperature of 68 °C [4,5,6]. Upon the MIPT, the crystalline structure changes from a high-symmetry tetragonal rutile phase (P4_2_/mnm, R phase) to a lower symmetry monoclinic phase (P2_1_/c, M_1_ phase) with dimerized V–V pairs exhibiting alternating zig-zag like chains [4,5]. Noticeable physical property changes such as optical transmittance [7,8,9], electrical conductivity [10,11], permittivity [12,13], and magnetic susceptibility [14,15] can be observed, making these kind of materials potentially suitable for a great many stimuli-responsive devices, e.g., ultrafast optical switches [16,17], sensing devices [18,19], next-generation terahertz transistors [20], non-Boolean computing materials [21], uncooled infrared bolometer [22,23], and other novel concepts of switching devices [24,25,26,27,28].

For device-based applications of VO_2_ materials [29], integration methods for film techniques are quite imperative, having the advantages of excellent purity, uniform thickness, feasibility in layer stacking, and are also compatible with large industrial level production. Many constructive methods for film deposition, ranging over both vapor and solution based techniques including sputtering [30,31,32], physical vapor deposition (PVD) [33], chemical vapor deposition (CVD) [34,35], pulsed laser deposition (PLD), sol-gel [36,37], and polymer-assisted deposition (PAD) [38,39] have been developed for VO_2_ films, which have shown excellent controls over the crystalline phases and chemical compositions. Recently, wafer-scale processing with superior thickness uniformity and valence state control have been demonstrated by rf magnetron sputtering [40] and a combinatorial approach [1], giving rise to constructive bridges to further fill the gaps between fundamental research and commercial devices. However, among the numerous reports on the synthesis of VO_2_ films, only a few distinct surface structures, e.g., rough pores [41], subwavelength nanoholes [42], nanotetrapods [43], nanobeams [19], have been developed. This may derive from the complexity of the vanadium oxide system [44] which has multiple oxidation states (from +2, as in VO, to +5, as in V_2_O_5_), different elemental compositions (such as V_n_O_2n−1_ and V_n_O_2n+1_) [45,46], and various coordination polyhedra (including the tetrahedron, trigonal bipyramid, square pyramid, regular octahedron, and distorted octahedron) [47]; however, these severely restrict their applications as high performance devices. Thus, the development of an effective strategy for the synthesis of VO_2_ films with desirable surface structures is still urgently required.

In this article, we reported on a novel wafer-scale oxygen-etching strategy that could successfully transform plain planar VO_2_ thin films into a highly textured nanopyramidal surface structure. The oxygen atmosphere applied in the experiments provides a mild environment (within the tolerance and stability) for VO_2_ chemical composition, which prevents further oxidizing into high valence vanadium oxides (e.g., V_2_O_5_, V_6_O_13_), although their surface structures change significantly from relatively smooth morphologies into nanopyramidal structures. Moreover, aside from the monoclinic M1 phase VO_2_, another metastable M2 phase, which is most frequently observed at high temperature during the metal-to-insulator phase transition, was also found in this investigation, even at room temperature. A temperature responsive field emission device was developed to broaden the applications of this special nanostructure. For a 3-inch wafer-scale VO_2_ film sample directly grown on silicon wafer, an increase of about three orders in the emission currency was observed after the oxygen-etching treatment, and the field enhancement factor β reached 3810 and 1620 for the M and R states, respectively, which are comparable with the reported ZnO nanobullets [48] and CdS nanobelts [49]. The simple and large-area available processing reported in this work may provide a practical strategy for the production of VO_2_-based integration devices.

## 2. Materials and Methods

### 2.1. Materials

Vanadium dioxide thin films were synthesized via a solution-based process developed previously in our laboratory [7,38]. Vanadium pentoxide (V_2_O_5_, analytically pure), polyvinylpyrrolidone (PVP, K90, average molecular weight 10,300,000) and diamide hydrochloride (N_2_H_4_·HCl, analytically pure) were used as the starting materials to form a 0.2 M vanadium precursor. Silicon pre-treated sequentially with ethyl alcohol, HCl, and NH_3_∙H_2_O were used as the deposition substrates. The precursor films were prepared by spin-coating at 1000 rpm for 30 s and dried at 60 °C for 10 min, and then were annealed in a N_2_ atmosphere at 520 °C for 1 h to obtain pure VO_2_ films (M1/R).

The oxygen-etching processing was performed in a quartz tube furnace (120 cm in length with a heating zone of around 20 cm in the middle). A pressure gauge was used to monitor the pressure inside the quartz tube, and two valves were installed at the gas inlet and outlet ends. A scheme for the furnace tube is shown in Scheme 1. For the oxygen-etching growth process, the above VO_2_ samples were loaded and rapidly heated to 540 °C in vacuum at a heating rate of 50 °C/min. The valve at the gas outlet end was then closed, and different amounts of the N_2_/O_2_ mixture (with an oxygen volume ratio of 2%) were introduced through the gas inlet. The content of oxygen introduced during annealing was monitored by the pressure changes recorded on the pressure gauge. The etching process was sustained for different spans of time, and textured vanadium oxide films with various morphologies were obtained. A schematic showing the evolution of the film morphology is given in Scheme 2.

### 2.2. Characterization

The surface morphologies of the films were determined by field emission scanning electron microscopy (SEM, JSM 6700F, JEOL, Tokyo, Japan). TEM images were acquired by transmission electron microscopy (TEM, JEM2010, JEOL, Tokyo, Japan). The samples for TEM were prepared by scraping the films from the substrates using a stainless steel blade; the resulting material was then dispersed in ethanol. X-ray diffraction was performed on a D/max 2550V X-ray diffractometer (Rigaku, Tokyo, Japan, Cu Kα, λ = 0.15406 nm). The film thickness was determined using a Taylor–Hobson surface profile measuring system by measuring at least four different points per sample. The structures and compositions of the samples were characterized by Raman spectroscopy and performed on a Raman microscope spectrometer (Raman, inVia Reflex, Renishaw, Gloucestershire, England) using a 514.5 nm laser.

To measure the field emission (FE) performance (as a cathode), the as-obtained nanofilms were placed before phosphor/indium tin oxide (ITO)/glass (anode), separated by two Teflon spacers with a thickness of 100 μm. The FE properties were measured under a high vacuum level of approximately 5 × 10^−5^ Pa at room temperature and 100 °C, respectively. The measured emission area was 1 × 1 cm^2^.

## 3. Results and Discussion

The as-prepared VO_2_ thin films before and after oxygen-etching treatment were characterized by room temperature XRD measurement (Figure 1). All of the recorded diffraction peaks for the original films obtained at 540 °C in a N_2_ atmosphere can be assigned to the pure monoclinic phase of VO_2_ (indexed as VO_2_ (M1), JCPDS card no. 72-0514, P2_1_/c, a = 0.57 nm, b = 0.45 nm, c = 0.54 nm, and β = 122.61°), in agreement with previous investigations [50,51]. However, after the introduction of oxygen with a controllable amount (0.12 kPa, sample labeled as S-0.12k in the following discussion) during the etching process, a slightly shifted diffraction peak of M1 VO_2_ (011) (from 27.86° to about 28.01°) and an extra peak at 28.53° were observed. The diffraction pattern at approximately 28.53° could not be assigned directly to any of the vanadium oxide phases, although it was close to the main diffraction peak of the (201) crystalline plane for M2 phase VO_2_ (JCPDS card no. 33-1441, space group C2/m, a = 0.908, b = 0.576, c = 0.453, and β = 91.3°). The crystallinity increased after the oxygen treatment, as deduced from the decreased full widths at half maximum (FWHM), which were 0.43° and 0.20° for the major diffraction peaks of the plain nanofilm and nanopyramid film, respectively.

The structures and compositions of the films before and after oxygen injection were further confirmed by room-temperature Raman spectra (Figure 2). For the original VO_2_ films, Raman modes corresponding to M1 phase VO_2_ were found, with peaks centered at 192, 223, 307, 338, 392, 440, 494, 612, and 816 cm^−1^ [51,52], respectively. However, after oxygen injection during post-annealing at 540 °C, other modes centered at approximately 271, 432, and 646 cm^−1^ appeared aside from the modes of the M1 phase of VO_2_. These modes can be assigned to the metastable M2 phase of VO_2_, whose Raman modes centered at around 226, 272, 293, 338, 432, 495, and 645 cm^−1^ [53]. The existence of these Raman modes, especially the characteristic V–V vibration mode at 646 cm^−1^, indicates that treating the VO_2_ films with a controllable amount of oxygen produced the VO_2_ M_1_/M_2_ composite structures, agreeing with the XRD results.

XPS spectra (Figure 3) were further conducted to investigate the compositions and valence states of the vanadium in the films. Surface contamination was removed before the measurements by an Ar ion (1 keV) etching treatment. As shown in a wide-range survey XPS spectrum, a typical prepared sample mainly consisted of vanadium and oxygen, besides the Si signals from the silicon substrate and carbon from the oil residues during pumping by a diffusion pump. The high-resolution XPS (HRXPS) profile of V2p shows the following two major valence states for the vanadium in the pyramidal VO_2_ films: V^4+^ with a binding energy of 516.0 eV and V^5+^ with a binding energy of 517.2 eV [52,54]. The quantitative analysis by peak fitting indicates a composition of VO_2.10_ for the pyramidal films, which is evidence of the oxidization of the surface of VO_2_ during the oxygen etching treatment.

To investigate the influence of oxygen on the morphology evolution of the pyramidal-like VO_2_ films, FE-SEM observations (Figure 4) were undertaken for the samples treated with oxygen for different times. From the SEM images, one can see that the growth of VO_2_ nanopyramids in the O_2_/N_2_ atmosphere unexpectedly followed the routes of particle top-surface sharpening, size enlarging, and then film structure loosening. Moreover, the growth rate for the oxygen treated sample was much higher than that of the sample annealed in a N_2_ atmosphere (see the dotted lines in Figure 4d). The particle sizes measured were 71 nm, 110 nm, and 162 nm for the samples treated in oxygen-containing environments for 5 min, 15 min, and 60 min, respectively; while the particle sizes of the samples obtained by annealing in a pure N_2_ atmosphere were only tens of nanometers. In fact, our further experiments confirmed that for samples treated without oxygen injection (in a N_2_ atmosphere), significant particle growth could only be observed when annealing at higher temperatures (e.g., above 700 °C), indicating that a new growth pattern was involved in the growth of nanopyramidal VO_2_ films.

The structure and composition of the pyramid films were further characterized by TEM, high resolution TEM (HRTEM), and fast Fourier transformation (FFT), and the typical results are given in Figure 5. As shown in the low magnification TEM image (Figure 5a), the pyramidal-like morphology for one individual particle can be deduced from the contour stripes. The FFT pattern (Figure 5c) for the total area of Figure 5b indicates that the major structure of the particle is M1 phase VO_2_ (in the direction of [010]). However, when the FFT patterns were performed in selected small regions, from the outer side to the inner one of the particle (as shown in Figure 5e–h), patterns belonging to M2 phase VO_2_ (in the direction of [200]) gradually appeared and became stronger and stronger. These results agreed well with the XRD and Raman results, further confirming the formation of composite phases in the pyramidal films.

As reported by others [53,55,56], the M2 phase is another monoclinic phase of VO_2_ with a free energy close to that of M1, which can be stabilized by doping [55] or compressive uniaxial stress in the R-[110] direction [56] (equivalent to a tensile strain along the *c* axis in R phase or *a* axis in the M1 phase). Considering the XRD results (Figure 1) where the diffraction patterns of M1 phase VO_2_ (011) (or R phase VO_2_ (110)) shifted toward high angles, a compressing stress along the R-[110] (or M1-[011]) direction can be expected, which will be favorable for stabilizing the metastable M2 phase. Moreover, the M2 phase has also been reported to be related with the incorporation of V^5+^ ions in the crystal lattices (by lower valence Cr^3+^ doping [55], or non-stoichiometric engineering caused by annealing in an oxygen-rich condition [57]). From the high-resolution TEM image in Figure 5b, we could see a distinct thin transition layer on the surface of the pyramidical VO_2_ nanoparticle, which had a lower contrast and less rigid lattice fringes when compared with the inside VO_2_ crystals, and the FFT pattern for this layer indicated the structure of M1 phase VO_2_. Considering the special oxygen treatment and the XPS results, this thin layer, especially for the outer parts, should be composed of slightly oxidized vanadium-oxide species, which works as an intermediate layer between the oxygen environment and the inside VO_2_ crystals in the annealing process.

When oxygen is injected into the vacuum tube, the top surfaces of the VO_2_ nanoparticles are expected initially to react with the excess O_2_ to form oxygen-rich vanadium oxides. The most far-gone form of these oxides is expected to be the highest valence state oxide of V_2_O_5_ (*T*_m_, V_2_O_5_ = 690 °C), which is highly volatile and can easily evaporate from the surface of the nanoparticles at the annealing temperature of 540 °C (the reaction could be expressed as VO_2_ (s) + O_2_ → V_2_O_5_ (s/l) → V_2_O_5_ (g)). In fact, evaporated V_2_O_5_ residues with a characteristic orange color can even be observed at the cold end of the quartz tube. The samples obtained with a higher oxygen injection content in our further experiments also confirmed the existence of V_2_O_5_ crystals surrounding the VO_2_ particles (to be discussed in the following section). An oxygen-related selective etching growth model can be described for the formation of a pyramidal film morphology.

As shown in Scheme 3, when oxygen is injected at a high temperature, the top surfaces of the VO_2_ nanoparticles react immediately with the excess oxygen to form a thin layer of oxygen-rich vanadium oxides, and the ultimate oxide of V_2_O_5_ in the thin layers evaporates partly into the gas phase, which causes the etching of the films. More importantly, due to the diversity in the active energies for different facets of VO_2_, when the oxygen is injected in a controlled low amount (as discussed for the S-0.12k sample), the reaction speed for different facets will vary. Specifically, because the monoclinic facet (011) is the lowest energy facet in VO_2_, the other reactive facets may react (or etch) preferentially with O_2_ to form volatile V_2_O_5_ and leave the particle surface. This can cause the preservation of VO_2_ (011) after etching, which agrees well with the observation that the (011) M1 orientation is highly preferred after oxygen etching.

However, for samples with higher oxygen injection contents, the above selective etching patterns were gradually eliminated. Figure 6 shows the SEM and TEM images for the samples treated with higher oxygen injection amounts of 0.20 kPa and 0.24 kPa (labeled as S-0.20k and S-0.24k, respectively). We can see from the SEM images that the top surfaces became much smoother for the S-0.20k film, and even evolved into a spherical (or semi-spherical) morphology for the S-0.24k film, indicating a significant suppression over the facet selective etching growth pattern. Furthermore, for the S-0.24k film, the spherical particles showed a distinct melted droplet-like morphology at the bottom of the etched particles, and large amounts of these particles showed a neck-connected structure (in the inset of Figure 6b).

TEM and HRTEM images were further taken to investigate the detailed structures of these films, and the results are given in Figure 6. As shown in Figure 6c,d, the thin transition layers that implied a surface oxidization process could be clearly found on the top surface of the particles for the S-0.20k film, although their lattice fringes were distorted more severely, indicating a much deeper oxidization level and more severe lattice defects with a higher oxygen injection content. For the S-0.24k film, the two distinct structures (in Figure 6b), e.g., the upside nanoparticles and the bottom-laying molten droplets, specifically, were also distinguishable in the TEM images (Figure 6e), and were assigned to the M1 phase VO_2_ and orthorhombic V_2_O_5_ (JCPDS Card File No. 41-1426) by their corresponding selected area electron diffraction patterns (the insets of Figure 6e). Furthermore, the top surface of the connected particles in Figure 6b and its inset were also confirmed as VO_2_ twin crystals with a M1 phase. Further investigation by XRD characterization indicated that a strongly orientated characteristic along the [011] crystallographic orientation of M1 phase VO_2_ could be observed for both films, which agreed well with the proposed selective etching mechanism of preserving the stable (011) crystal facets while consuming others during the oxygen treatment. Moreover, when compared with the nanopyramidal films, both films exhibited a smaller FWHM for the (011) diffraction peak, recorded with 0.11° and 0.17° for S-0.20k and S-0.24k, respectively, indicating an increased crystallinity or prompted growth process in this orientation during annealing. However, for the S-0.24k sample, the oxygen concentration applied at 540 °C was too high to stabilize the VO_2_ crystals, and large amounts were oxidized to be orthorhombic V_2_O_5_, which can be seen from the XRD results in Figure 7, in accordance with the TEM observations of the bottom melted droplet-like compounds in Figure 6e.

All of the above results indicate that the formation of VO_2_ nanopyramidal structures represents the interplay of the thermodynamic stability of different vanadium oxide phases and the kinetics of vanadium oxidation. With a low oxygen injection content (as discussed for the S-0.12k sample), the reactive facets located on the surfaces of pristine VO_2_ crystals react immediately with O_2_ to form thin oxygen rich transition layers, although some of these layers are gradually oxidized into highly volatile oxides of V_2_O_5_, resulting in a selective etching growth pattern for the nanopyramidal structures. When a higher content of oxygen was injected (as discussed for the S-0.20k sample), the selective etching process may have been suppressed to a certain extent because of the rapid reaction with a large amount of surrounding oxygen, which could be deduced from the gradually smoothened particle morphology. Furthermore, when the applied oxygen content was too high (as discussed for the S-0.24k sample), large amounts of V_2_O_5_, in the vapor phase and/or melted droplet states at the temperature of 540 °C, could be formed aside from the VO_2_ crystals. The growth patterns for this kind of nanostructure were followed, partly with the routes of oxygen related etching growth as we proposed, although a liquid involved growth mechanism [58,59] should also be considered. After all, by simply controlling the oxygen injection content as discussed above, highly textured VO_2_ films, especially the nanopyramidal-like nanostructures that have rarely been reported, can be formed, indicating that the current method by oxygen relation etching growth is of competitive potential for the synthesis of nanostructured VO_2_ films.

Moreover, to explore potential applications of the oxygen etching treated nanostructured VO_2_ films, field emission (FE) devices were assembled and investigated. The FE device is one of the key features by which nanostructured materials can eject electrons from their surfaces into the vacuum energy level under high electric fields [60,61], and useful applications such as high performance X-ray sources [62,63] can be produced based on the FE properties. According to reports from Yin and Yu [64], VO_2_ nanobundles exhibit a high field enhancement factor (β) of 1020–1400, which is better than that of Si [65] and Ge nanowires [66], and is comparable to ZnO nanobullets [48] and CdS nanobelts [49]. Furthermore, the Fermi level of VO_2_ increases distinctly due to the metal-to-insulator phase transition, resulting in a rapid decrease in the work function [64], which makes VO_2_ materials quite suitable in applications such as temperature-regulated field emission devices. Comparatively, nanopyramidal-like VO_2_ nanostructures present promising sharp tips that can facilitate electron emission, so a higher field emission performance can be expected.

The relationship of the field emission current density J with the applied electric field E (J‒E curves) for the device based on the VO_2_ thin films before and after the oxygen-etching treatment (corresponding to sample S-0.12k) is shown in Figure 8. The FE measurements were performed at both 25 and 100 °C, respectively. A schematic of the field emission measurement is given in Figure 8a, and a photograph of a large area of VO_2_ nanopyramidal film (diameter of ~7.6 cm) obtained by this method is given in Figure 8b. As shown in the J‒E curves (Figure 8c,d), the device based on the original VO_2_ film (the dark yellow lines) showed no or undetectable FE properties at either low or high temperature, while the device made with the nanopyramidal VO_2_ film after oxygen-etching treatment showed a highly enhanced emission currency at both low and high temperatures, showing an exponential profile versus the applied electric fields. The emission currencies measured at a fixed electric field of 6 V/μm were 1.6 × 10^−6^ A/cm^2^ (or 7.2 × 10^−7^ A/cm^2^) and 1.6 × 10^−3^ A/cm^2^ (or 1.9 × 10^−5^ A/cm^2^) at the R (or M) state before and after the oxygen-etching treatment, indicating an improvement of nearly 1000 times (or 26 times) in emission currency at the R (or M) state. The increase in emission currency can be expected to originate from the significant changes in film particle morphologies, which can pare away the difficulties of electron field emission for sharp-tipped pyramidal surfaces. Furthermore, the work function of the M1 phase was 0.07 eV, higher than that of the R phase [64], so its improvement in the field emission property was much more significant in the R state after transforming from plain thin films with no FE properties.

The Fowler‒Nordheim (F‒N) plots of the ln(J/E^2^)‒1/E relationship [61] are also shown in the insets of Figure 8c,d. The straight lines fitted in the F‒N plots confirmed the electron field emission characters for the pyramidal VO_2_ film, which could not be observed for the original plain VO_2_ thin film. By determining the slop of the F–N plot, one can obtain the data of the field enhancement factor (β), estimated to be about 3810 and 1620 for the M and R states, respectively, which were much higher than those of the reported VO_2_ nanobundles [64], and comparable with high quality ZnO nanoneedles [67], making the pyramidal VO_2_ films prepared by the present method attractive for FE applications. Moreover, the feasible and large-area available synthetic strategy reported here can also provide an efficient tool to build a variety of interesting nanoscale VO_2_ surfaces, which is beneficial for VO_2_ and other transition metal oxide based device applications.

## 4. Conclusions

In this paper, we report on an effective wafer-scale oxygen-assisted etching growth strategy for transforming plain VO_2_ thin films into highly textured nanopyramidal thin films. The controllable amount of oxygen incorporated during the post annealing process reacted rapidly with the reactive VO_2_ planar surface to form volatile V_2_O_5_, which resulted in the effective etching and formation of pyramidal-like VO_2_ films. The structural investigation indicated that the VO_2_ films had a composite M1/M2 structure at room temperature. Field emission measurements further showed that the etching process could successfully transform planar VO_2_ films with no field emission performance into nanopyramidal films with significant field emission performance. The emission currencies at a fixed electric field of 6 V/μm could be improved by 1000 times at the R state before and after the oxygen etching treatment, and the field enhancement factor β reached 3810 and 1620 for the M and R states, respectively, which was quite high amongst all reports. All these results indicate that the technique developed here is an efficient surface building and reconstruction strategy for VO_2_-based thin films with potential applications in high-performance FE devices.

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
