# Peer review of "Large Scale Synthesis of Nanopyramidal-Like VO2 Films by an Oxygen-Assisted Etching Growth Method with Significantly Enhanced Field Emission Properties"

_nanomaterials, 2019, doi:10.3390/nano9040549_

Round 1
Reviewer 1 Report
FE characteristics are not great, but the methods and attempt is good with oxygen-assisted etching growth. There are too many wrong text in manuscript, please refer to attached file.
(line #142) There are not mentioned about 271, 432 in the Fig. 2. Please add the information in the Fig. 2.
(Fig. 4) Three are not mentioned inset figure. 4. Please explain about inset image.
(line #195) For the reader who does not know [110] of crystal result, please express the information differently. It is not distinguishable from reference display.
(Fig. 8) Please add the information text in graph.
(Fig. 8) Please change the "mA" to "A" in Y-axis. also, it is not appropriate to use the prefix (m in mA) and the 10 ^ 4 representation in manuscript (#311).
(line #48) Please introduce the references to latest X-ray applications in field emission.
1. Gupta, A. P.; Park, S.; Yeo, S. J.; Jung, J.; Cho, C.; Paik, S. H.; Park, H.; Cho, Y. C.; Kim, S. H.; Shin, J. H.; Ahn, J. S.; Ryu, J. Direct Synthesis of Carbon Nanotube Field Emitters on Metal Substrate for Open-Type X-ray Source in Medical Imaging. Materials 2017, 10, 878.
2. Park, S.; Gupta, A. P.; Yeo, S. J.; Jung, J.; Paik, S. H.; Mativenga, M.; Kim, S. H.; Shin, J. H.; Ahn, J. S.; Ryu, J. Carbon Nanotube Field Emitters Synthesized on Metal Alloy Substrate by PECVD for Customized Compact Field Emission Devices to Be Used in X-Ray Source Applications. Nanomaterials 2018, 8.

Author Response
The authors feel grateful for your kind and helpful suggestions. We have revised our manuscript accordingly, and the corresponding revisions are as follows:
(1) (line #142) There are not mentioned about 271, 432 in the Fig. 2. Please add the information in the Fig. 2.
Response: Thanks for your suggestion. Raman modes of “271” and “432” were added and marked properly in Fig.2.
(2) (Fig. 4) Three are not mentioned inset figure. 4. Please explain about inset image.
Response: Thanks for your suggestion. We have added the corresponding information for the insets in Fig.4.
In the last part of figure caption (line # 179), a sentence of "The inserts in figure (4a) and (4b) showed the high magnification surface images for the corresponding sample, and the insert in figure (4c) showed the enlarged cross-section image of the sample.” was added.
(3) (line #195) For the reader who does not know [110] of crystal result, please express the information differently. It is not distinguishable from reference display.
Response: Thanks for your kind suggestion. The expression in line 195 was changed accordingly.
In line 195, the sentence of “… or compressive uniaxial stress in the [110] direction of the R phase...” was changed to “… or compressive uniaxial stress in the R-[110] direction...”
(4) (Fig. 8) Please add the information text in graph.
Response: Thanks for your suggestion. The information for the two different samples were added in Fig. 8.
(5) (Fig. 8) Please change the "mA" to "A" in Y-axis. also, it is not appropriate to use the prefix (m in mA) and the 10 ^ 4 representation in manuscript (#311).
Response: Thanks very much for your kind suggestion. Corresponding changes were made in Fig.8, and a revision of the expression was also given in the manuscript.
In line #311, the sentence of “The emission currencies measured at a fixed electric field of 6 V/μm are 1.6×10-3 A/cm2 (or 7.2×10-4 A/cm2) and 1.6 mA/cm2 (or 1.9×10-2 A/cm2)...” was changed to “The emission currencies measured at a fixed electric field of 6 V/μm are 1.6×10-6 A/cm2 (or 7.2×10-7 A/cm2) and 1.6×10-3 A/cm2 (or 1.9×10-5 A/cm2)...”
(6) (line #48) Please introduce the references to latest X-ray applications in field emission.
Response: Thanks for your suggestion. The two references indicating the latest advances of X-ray applications in field emission were added.
In the revised manuscript, a sentence of “…, and useful applications such as high performance X-ray sources [62, 63] can be produced based on the FE properties.” was added. (Because the parts around line # 48 mainly show the applications of VO2 based stimuli-responsive devices, after a consideration, we added these two useful references in line #293 to describe the advancing research results in FE applications)
In the reference part, the following to references were added, and the sequences of the reference part were revised accordingly.
62. A. P. Gupta, S. Park, S. J. Yeo, J. Jung, C. Cho, S. H. Paik, H. Park, Y. C. Cho, S. H. Kim, J. H. Shin, J. S. Ahn, J. Ryu, Direct synthesis of carbon nanotube field emitters on metal substrate for open-type X-ray source in medical imaging. Materials, 10 (2017) 878.
63. S. Park, A. P. Gupta, S. J. Yeo, J. Jung, S. H. Paik, M. Mativenga, S. H. Kim, J. H. Shin, J. S. Ahn, J. Ryu, Carbon nanotube field emitters synthesized on metal alloy substrate by PECVD for customized compact field emission devices to be used in X-ray source applications. Nanomaterials, 8 (2018) 378.
Reviewer 2 Report
I strongly recommend the publication of this valuable paper. I suggest a revision of editing to enhance the quality of the manuscript. I also would like to suggest:
- In Fig 2 and Fig 7a please use the scientific notation for the Intensity scale, like in Fig 3a
- In line 194 you say "As reported by others...", please, add references.
Author Response
The authors feel grateful for your kind and helpful suggestions. We have revised our paper accordingly, and the corresponding revisions are as follows:
(1) In Fig 2 and Fig 7a please use the scientific notation for the Intensity scale, like in Fig 3a.
Response: Thanks for your suggestion. Fig 2 and Fig 7a were revised accordingly, and the corresponding changes can be seen in the revised manuscript.
(2) In line 194 you say "As reported by others...", please, add references.
Response: Thanks for your suggestion. The corresponding references of [53], [55] and [56] were added.
Author Response
Thanks very much for your highly positive comments on our work.
We have made some further small revisions on the inappropriate expressions or styles to improve the quality, which can be seen in the submitted revised manuscript (marked in red).
Thanks again.